# New Epoxy Thermosets Organic-Inorganic Hybrid Nanomaterials Derived from Imidazolium Ionic Liquid Monomers and POSS^®Ph^

**DOI:** 10.3390/nano12030550

**Published:** 2022-02-06

**Authors:** Houssém Chabane, Sébastien Livi, Jannick Duchet-Rumeau, Jean-François Gérard

**Affiliations:** 1Univ Lyon, CNRS, UMR 5223, Ingénierie des Matériaux Polymères, Université Claude Bernard Lyon 1, INSA Lyon, Université Jean Monnet, F-69621 Villeurbanne, France; houssem.chabane@insa-lyon.fr (H.C.); sebastien.livi@insa-lyon.fr (S.L.); jannick.duchet@insa-lyon.fr (J.D.-R.); 2Laboratoire de Chimie Macromoléculaire, Ecole Militaire Polytechnique, BP 17, Bordj El-Bahri, 16111 Algiers, Algeria

**Keywords:** diepoxydized ionic liquid, polyhedral oligomeric silsesquioxane (POSS^®^), epoxy networks, ionic conductivity

## Abstract

New epoxy-amine networks issue from epoxydized imidazolium ionic liquid monomers (ILMs) and isophorone diamine (IPD) were modified for the first time by incorporating unmodified trisilanol phenyl POSS^®^ (POSS^®Ph^-triol) and two ionic liquid-modified POSS^®Ph^ (IL-g-POSS^®Ph^) having chloride (Cl^−^) and bis-trifluoromethanesulfonimidate (NTf_2_^−^) counter anions. Then, 5 wt.% of unmodified and IL-modified POSS^®Ph^ were introduced in order to develop new solid electrolytes. First, a homogeneous dispersion of the POSS^®Ph^ aggregates (diameters from 80 to 400 nm) into epoxy networks was observed. As a consequence, ILM/IPD networks with glass transition temperatures between 45 and 71 °C combined with an enhancement of the thermal stability (>380 °C) were prepared. Moreover, a significant increase of the hydrophobic character and high oil repellency of the network surfaces were obtained by using IL-g-POSS^®Ph^ (19–20 mJ.m^−2^), opening up promising prospects for surface coating applications. Finally, these new epoxy networks exhibited outstanding high ionic conductivities (from 3.4 × 10^−8^ to 6.8 × 10^−2^ S.m^−1^) combined with an increase in permitivity.

## 1. Introduction

In recent years, numerous works have reported on the design of new electrolytes in order to propose alternatives to liquid electrolytes containing organic solvents, i.e., to ensure the safety of the installations, avoiding leakage of liquids and related explosions [1,2,3,4]. Among these, two strategies were proposed focusing on the development of polymer electrolytes including solid polymer electrolytes (SPEs) and gel polymer electrolytes (GPEs) [4,5]. Solid polymer electrolytes (SPEs) are considered as the best alternative to solve safety issues, combined with an improvement of the mechanical behavior and capability of holding liquid electrolytes, as well as to obtain high ionic conductivity with good electrochemical stability [6,7]. However, there is still work to be undertaken before finding ideal solid polymer electrolytes that respect all the parameters required at the same time—i.e., high ionic conductivity, excellent electrochemical stability, low interfacial resistance between electrode and electrolyte, good mechanical integrity, high thermal stability, etc. These requirements have not been concurrently fulfilled to enable practical electrolyte applications [8,9]. Various SPEs combined with high Li^+^ conductivity have been developed [6,10,11,12]. Polyethylene oxide (PEO) incorporated with Li salts [13] was the first SPE reported in the literature, where the transport of Li^+^ ions was affected by the number of free Li^+^ ions and their displacement capacity. Yuan et al. [14] reported a high ionic conductivity of 3.16 × 10^−5^ S cm^−1^ at 25 °C by incorporation of Zn-based MOF-5 into PEO-based composite polymer electrolytes using in situ method [14]. Gerbaldi et al. [15] also achieved high performance SPE for Li-based batteries by using aluminum-based MOFs (Al-BTC MOFs) dispersed in a PEO matrix [15]. However, despite their high ionic conductivity value, these SPEs have many disadvantages such as lower mechanical properties, a reduced electrochemical stability window, a poor charge-discharge stability and unfavorable interface contacts with the electrodes. In addition, their difficult processing conditions considerably limit their applications [16,17].

The main advantages of polyepoxides for electrical applications are their low relative permittivity (typically three to six), their low losses and dissipation, their excellent dielectric strength (can reach 180 kV/mm in DC) as well as their low conductivity (from 10^−12^ to 10^−19^ S/m). As a result, polyepoxides are widely used and studied as insulators, and only few works have reported the conduction mechanisms of these systems. On the other hand, the conduction mechanisms have been widely investigated in the field of ionic conductive polymers, namely electrolyte polymers. Due to their good chemical stability and high ionic conductivity, ionic liquids are good candidates [16,17]. Ionic liquids (ILs) are salts composed of ammonium, imidazolium or even phosphonium cations, and combined with organic or inorganic anions with a low melting temperature. One of their advantages is their great versatility that can induce good chemical and thermal stabilities, as well as significant ionic conductivity. These properties make them, for example, interesting as a lubricant, electrolyte or additive in polymer science [12,13,14,15,16,17]. The association of ionic liquids with polyepoxides has been studied in the literature for use as an initiator and/or catalyst for the polymerization of epoxies, but also as a solution for electrolyte applications. Polymerized ionic liquids (PILs) combine both the properties of ILs (wide electrochemical window, non-volatility, and tunable molecular design) and of polymers (avoiding liquid leakage, no need of a separator, mechanical performances, and easier implementation) [18]. Recently, PILs prepared by polyaddition have been preferred for preparing SPEs to get over the challenges and limitations of conventional two-ion conductive SPEs [12,13,14,15,16,17]. Various PILs based on imidazolium have been considered in the literature [18,19,20,21]. The majority of these studies involved mono-cationic PIL-based electrolytes displaying significant ionic conductivities by using large contents of ionic liquids (from 10 to 90 wt.%) for temperatures ranging from 50 to 100 °C [22,23,24,25]. For example, Obadia et al. [23] have prepared a series of anionic poly(ionic liquid)s with 1,2,3-triazolium counter cations by cation exchange between tailormade 1,3,4-trialkylated-1,2,3-triazolium iodides and a polystyrene derivative having as side chain potassium groups bis-(trifluoromethylsulfonyl)imide ones. The resulting PILs networks showed a great potential for applications that require solid electrolytes with ionic conductivities from 7.8 × 10^−8^ to 8.4 × 10^−7^ S.m^−1^ at 30 °C [22,23]. Other authors proposed materials having an ionic conductivity of 8.6 × 10^−9^ S.m^−1^ at 30 °C by combining 10 wt.% of IL electrolyte with poly(diallyldimethylammonium) bis-(trifluoromethanesulfonyl)imide (PDADMA TFSI) [26,27]. Ionic conductivity higher than 10^−5^ S.m^−1^ at the same temperature was obtained by increasing the IL content up to 40 wt.% [25]. Recently, Porcarelli et al. [24] synthesized different types of methacrylic ILMs monomers copolymerized with poly(ethylene glycol) methyl ether methacrylate (PEGM) via conventional free radical reaction. The resulting solid polymer electrolytes showed high ionic conductivities (up to 1.9 × 10^−6^ and 2 × 10^−5^ S.m^−1^ at 25 and 70 °C, respectively) combined with a low T_g_ (from −61 to −27 °C) depending on copolymer composition and ILM/PEGM ratio [24]. Solid polymer electrolytes having a T_g_ close to room temperature showed a very low conductivity (lower than 10^−11^ S.m^−1^) [24]. Livi et al. [28] obtained (multi)functional epoxy ionic liquid networks from the copolymerization between an imidazolium ionic liquid monomer and conventional polyether-amine (Jeffamine D230) which has a high ionic conductivity of about 7.39 × 10^−8^ and 4 × 10^−4^ S.m^−1^ at 30 and 70 °C, respectively.

Very recently, POSS^®^-based materials were described which can enhance the thermal and electrochemical stabilities of SPEs due to their unique structure and the fact that their inner inorganic silicon oxygen core facilitates ion conduction [29,30]. The incorporation of POSS^®^ nanoclusters in an epoxy-based matrix allows for the enhancement of oxidative and thermal resistances [31,32,33], surface characters (such as adhesion and wettability) [34,35], combustion performances [33], and dielectric as well as mechanical properties [36,37]. POSS^®^ nanoclusters can be used for designing solid polymer electrolytes for lithium batteries and fuel cells [38,39,40]. Gyu et al. [29] described nanomaterials prepared from of polyethylene oxide (PEO)/polyethylene glycol-polyhedraoligomeric silsesquioxane (PEG-POSS^®^) combinations which exhibit enhanced ionic conductivity (from 1.19 × 10^−5^ to 1.27 × 10^−4^ S.cm^−1^). Lu et al. [30] also obtained crosslinked polymer membranes (3D-PEs) with star-shaped structures having ionic conductivity up to 2.35 × 10^−3^ S.cm^−1^ at room temperature with the introduction of a multifunctional epoxy POSS^®^. These authors also observed that the most of the POSS^®^-epoxy networks display higher real and imaginary parts of the complex permittivity compared to the ones of the neat epoxy network. Recently, some researchers have reported the grafting of ionic liquids onto octa-silsesquioxane (IL-g-POSS^®^) and investigated if they could be useful to prepare new solid electrolytes polymers [41,42]. The grafting of ionic liquids onto POSS^®^ [41,42] has shown a strong improvement of ionic conductivity, excellent electrochemical stability, and high thermal stability compared to systems which only contain POSS^®^ [38,39,40] or POSS^®^ with free ionic liquid [43]. Our previous studies reported also that the connection of ion pairs to POSS^®Ph^ with different organic ligands (isobutyl or phenyl) can significantly help to improve the O/I interphase and promote the formation of well-dispersed POSS^®^ nanodomains [44] leading to nanomaterials with outstanding thermal and mechanical properties [45]. From these previous results and the ones reported in the literature, it is plausible that a new generation of electrolyte could be designed. In addition, novel epoxy-functionalized mono-imidazolium or bis-imidazolium ionic liquid monomers have been synthesized in our group to substitute the bisphenol A diglycidyl ether prepolymer (DGEBA) as starting components [46,47]. Moreover, to the best of our knowledge, there is no work investigating the effect of IL grafted POSS^®^ introduction into epoxy PILs.

In the present work, a new generation of nanostructured epoxy PILs is prepared by combining POSS^®^ and ionic liquid monomers. For this, two types of epoxy ionic liquid monomers (ILM1 and ILM2) based on imidazolium salt have been synthesized as well as three types of POSS^®Ph^ O/I nanoclusters (POSS^®Ph^ -triol and IL-g-POSS^®Ph^ with different anions). First, the influence of these epoxy ionic liquid monomers structure on the dispersion morphology of POSS^®Ph^ was investigated based on transmission electron microscopy (TEM). Then, the thermal properties (DSC, TGA), surface properties, and the ionic conductivity behavior of these new hybrid O/I nanomaterials have been investigated and compared with conventional DGEBA-IPD networks.

## 2. Materials and Characterization Methods

### 2.1. Materials

All reagents purchased from Sigma Aldrich Co (St. Louis, MO, USA)., or TCI Co (Paris, France). were used as received. Solvents were used in RPE grade without further purification. Anhydrous solvents were obtained from a Puresolv SPS400 apparatus developed by Innovative Technology Inc. (Hong Kong, China) Epoxy prepolymer based on diglycidyl ether of bisphenol A (DGEBA), denoted Epon 828, was provided from Hexion Co (Louvain-la-Neuve, Belgium). The isophorone diamine (IPD), from Aldrich Co. (St. Louis, MO, USA), was used as hardener with epoxy ionic liquid monomers (ILM1 or ILM2), commercial epoxy prepolymer DGEBA. Heptaphenyl-trisilanol POSS^®Ph^, denoted POSS^®Ph^-triol, from Hybrid Plastics Co (Hattiesburg, MS, USA). was used as received. All the chemical formulae of components used are summarized in Table 1.

### 2.2. Epoxy Network Preparation

To prepare epoxy-amine networks, epoxy prepolymer (DGEBA) or epoxy ionic liquid monomers (ILM1 or ILM2) (see Table 1) were mixed with the isophoronediamine (IPD) used as co-monomer under mechanical stirring with a stoichiometric ratio amino hydrogen-to-epoxy equal to 1. For the POSS^®Ph^-modified epoxy networks, the 5 wt.% of POSS^®Ph^-triol or IL-g-POSS^®Ph^ were premixed in the DGEBA epoxy prepolymer or the epoxy ionic liquid monomers (ILM1 or ILM2) at 100 °C for 45 min in order to obtain a homogeneous and transparent solution before adding the isophoronediamine. Then, the mixture was degassed and poured into molds. Finally, the different systems were polymerized under different conditions: *i/* The neat epoxy-amine and the POSS^®Ph^-modified epoxy based on ILM2 networks were cured for 1 h at 140 °C and post-cured for 8 h at 190 °C whereas the other system based on ILM1 was cured for 3 h at 80 °C and 6 h at 120 °C, followed by a post-curing process at 200 °C for 1 h in order to ensure a complete crosslinking of the resulting networks.

### 2.3. Determination of the Soluble Fraction Obtained from Hybrid O/I Networks

The solubilization was considered to determine the POSS^®Ph^ fraction covalently bonded to the architecture of the networks. The extraction was undertaken using THF at 60 °C for 24 h and the THF was removed by evaporation using rotary evaporator. The soluble fraction, *w_s_*, was evaluated from the weight of the dry sample before, *m_dry_*, and after extraction (*m_dry-ext_*) using the following equation:w_s_ = 1 − (m_dry-ext_)/m_dry_)

### 2.4. Characterization Methods

Fourier Transform Infrared Absorption Spectra (FTIR) were recorded using a Thermo Scientific Nicolet iS10 spectrometer in attenuated total reflectance (ATR) mode from 4000 to 500 cm^−1^ (32 scans, resolution 4 cm^−1^) used to identify the soluble fraction obtained from the solubilization of the hybrid O/I networks in the THF. In addition, the reaction kinetics, i.e., the quantification of the epoxide groups conversion, for all the reactive systems were investigated by FT-IR spectroscopy following the curing conditions: 3 h at 80 °C, 6 h at 120 °C, and 1 h at 200 °C for ILM1/IPD and 1 h at 140 °C followed by 8 h at 190 °C for ILM2/IPD. All the reactive systems were analyzed in the wavelength region of 800–1300 cm^−1^ in order to follow the changes of the adsorption bands at 914 and 1132 cm^−1^ corresponding to the epoxy groups with reference as the absorption band at 1132 cm^−1^ (trifluoromethyl (-CF_3_) of bis(trifluoromethanesulfonyl)imidate counterions (NTf_2_^−^) [46], respectively, by using the following equation [28,45,46]:$$X\%=\frac{A_{0}-A_{t}}{A_{t}}\times 100\%$$
where *A*_0_ and *A_t_* are the ratios between the area of two absorption bands at 914 and 1132 cm^−1^ (*A*_914_/*A*_1132_) of the reactive system at the beginning of the reaction (*t* = 0) and at a given reaction time t. Thus, the epoxy conversion versus the polymerization time is presented in Appendix A.

^13^C HR-MAS NMR spectroscopy analyses were undertaken using a Bruker Advance II spectrometer (Wissembourg, France) (400 MHz) equipped with a 4 mm diameter rotor ^1^H-^13^C HR-MAS probe with z-gradient coil at 5 kHz rotation speed operating at 298 K.

Thermogravimetric analyses (TGA) of all the epoxy-IL and their hybrid O/I networks and were performed using a Q500 thermogravimetric analyzer (TA Co. Ltd., New Castle, DE, USA). The samples were heated from 30 to 900 °C at a heating rate of 20 K.min^−1^ under nitrogen and air flow.

Differential scanning calorimetry analyses (DSC) of ILMs reactive system and thermosetting networks were performed on a Q10 (TA Co. Ltd., New Castle, DE, USA) in a dynamic mode, i.e., with a heating rate of 10 K.min^−1^ under nitrogen flow of 50 mL.min^−1^ from −70 to 300 °C.

The morphology of the networks was evaluated by transmission electron microscopy (Philips CM120) (Waltham, MA, USA). Samples 60 nm thick were previously cut by ultramicrotomy and placed on a copper grid for analysis.

Surface energy of epoxy networks was determined according to the sessile drop method using a GBX goniometer (Dublin, Ireland). From contact angle measurements performed with water and methylene diiodide as probe liquids, nondispersive and dispersive components of surface energy were determined according to the Owens–Wendt theory [48].

Dielectric spectroscopy measurements were performed on an Ametek Solartron XM spectrometer (Elancourt, France). The measuring cell and cooling system were supplied by Janis Research Company. A sinusoidal voltage of ±5 V is applied to the terminals of the sample, for a frequency range defined from 10^6^ to 10^−1^ Hz. Two temperature sweeps from −80 to 200 °C were applied successively, with an isothermal measurement in frequency sweep every 3 °C. The electrodes used are made of brass and have a diameter of 25 mm. In order to ensure electrical contact, the samples are first metallized with gold. Dielectric measurements were undertaken under isothermal conditions (3K temperature steps) with a frequency range from 1 MHz to 0.1 Hz (10 points per decade) applying V_rms_ = 5 V. The complex conductivity, σ*, was calculated from the Equation (1):σ* (ω) = ω.*ε*_0_.*ε** (ω)(1)
where the angular frequency ω = 2πf (f being the frequency), *ε*_0_ the vacuum permittivity (8.85 × 10^−12^ F.m^−1^), and *ε** the complex permittivity of the material. The DC conductivity measurements were carried out at low temperature (T < T_g_). The σ_DC_, was extrapolated from the real part of the conductivity σ′ (ω) = ω.*ε*_0_.*ε*″ (ω) for ω → 0, where a plateau is observed.

## 3. Results and Discussion

### 3.1. Morphology of Hybrid O/I Networks

Considering phenyltrisilanol POSS^®^ (POSS^®Ph^-triol) or IL-g-POSS^®Ph^ combined with the various counter anions, i.e., chloride (Cl^−^) versus bistriflimide (NTf_2_^−^), homogeneous and transparent mixtures with DGEBA or ILMs and IPD have been observed. This phenomenon confirms the solubility of all the monomers at the initial stage, i.e., before polymerization. The miscibility could be associated with the formation of intermolecular interactions (e.g., including Van der Waals, electrostatic, solvophobic, steric, and hydrogen bonds interactions) between POSS^®Ph^ and ILMs or DGEBA (and/or IPD). These homogeneous and transparent mixtures based on 5 wt.% of non-modified (POSS^®Ph^-triol) or IL-modified POSS^®Ph^ (IL-g-POSS^®Ph^) were cured at elevated temperatures to prepare in situ hybrid organic-inorganic networks. It was observed during the curing reaction that for DGEBA prepolymer-based systems, the initially transparent solutions gradually became cloudy evidencing the occurrence of a phase separation. This phenomenon is induced by polymerization and is denoted RIPS (Reaction Induced Phase Separation). However, with ILMs, the resulting networks were homogeneous and transparent; i.e., no discernible phase separation occurred at micro/macroscale.

Transmission electron microscopy was considered to highlight the dispersion state of the POSS^®Ph^-triol and IL-g-POSS^®Ph^ nanoclusters into epoxy networks as well as the formation of a possible interphase between the epoxy ILM-based network and POSS^®Ph^-POSS^®Ph^ aggregates. The morphologies obtained of the epoxy hybrid O/I networks are displayed in Figure 1. In all TEM micrographs, the dark part corresponds to the POSS^®Ph^-rich phases whereas the brightest continuous phase corresponds to the epoxy ILM/IPD matrix.

In all cases, i.e., the epoxy networks based on ILM1 or ILM2 containing POSS^®Ph^-triol or IL-g-POSS^®Ph^, a homogeneous distribution of nanoclusters aggregates in the epoxy matrix was revealed by TEM. The main difference between the epoxy ILM/POSS^®Ph^-triol and ILM/IL-g-POSS^®Ph^ (Figure 1) hybrid networks is the average size of the POSS^®Ph^-POSS^®Ph^ aggregates. The POSS^®Ph^-rich dispersed phases have a spherical or ellipsoidal shape in the different systems, with a diameter range from 60 to 900 nm for the epoxy networks based on ILM1 or ILM2 epoxy monomers. Such features are usually encountered for conventional morphologies generated from a phase separation induced during polymerization [49,50,51]. In both epoxy networks (ILM1 or ILM2-based ones) containing POSS^®Ph^-triol, the formation of an interface is observed around some POSS^®Ph^ aggregates as shown in Figure 1a,d. This interface has an average thickness included between 10 and 30 nm measured directly from the TEM images by using Image J Software (ImageJ bundled with 64-bit Java 1.8.0_172, 2020). This interface represented about 2% of the total volume of the POSS aggregate. This interface can be also explained by the low amount of POSS^®Ph^-triol nanoclusters which reacted with the epoxy-amine matrix. The interconnected nanophase-separated morphology with uniform domain size is characteristic of self-assembled structures generated as result of a balance of different intermolecular and intercluster interactions [52,53]. It should be pointed out that the dispersion of POSS^®Ph^-triol clusters in the ILM1 or ILM2-based matrix is better than the one obtained with DGEBA/IPD network, where the diameter distribution is ranging from 0.1 to 1.5 µm [45]. For epoxy ILM/IPD networks containing IL-g-POSS^®Ph^ with chloride (Cl^−^) or bistrifluoromethanesulfonimidate (NTf_2_^−^) counter anions, the TEM micrographs show that the phase separation occurred at nanoscale. In fact, the POSS^®^-rich dispersed phase shows spherical shapes based on small assemblies of POSS^®Ph^, with an average diameter from 80 to 400 nm (Figure 1), which can be seen as the result of the nanostructuration of POSS^®Ph^ cages. In addition, the dispersion of IL-g-POSS^®Ph^ having a Cl^−^ anion in the ILM1 or ILM2 epoxy monomer-based networks is always better than the one for IL-g-POSS^®Ph^ having a NTf_2_^−^ anion (uniform state of dispersion and aggregates sizes). These differences can be explained by the higher nucleophilicity and the smaller molar mass of the chloride anion compared to the NTf_2_^−^ anion [54]. The Cl^−^ provides a higher mobility leading to a high reactivity with the ILMs monomers. According to the literature, various authors have explored the ability of imidazolium ionic liquid to be used as a dispersing agent or dispersion media [55]. In fact, these ones could facilitate the dispersion of metallic [56] or inorganic particles [57] as well as carbon nanotubes [58]. The average size and size distribution of nanoparticles are managed from the physico-chemical properties of ionic liquids. For example, stabilization of bare silica nanoparticles by imidazolium-based ionic liquids was found to be tuned by the affinity between the ions and particle surface according to careful selection of the ions of ILs and/or the nanoparticle surface modification. Such a management could optimize the ionic liquid-based steric stabilization provided by the non-polar alkyl chains as a protecting group [59]. Donato et al. have obtained well-dispersed silica aggregates (of about 30 nm) for 6.8 wt.% of IL-functionalized silica particles having different counter anions (Cl^−^ and MeSO_3_^−^) [60,61]. These obtained dispersions are finer than those using a metal complex catalyst, i.e., aluminum triacetylacetonate ([Al]) which is used to reduce the size of POSS^®Ph^-triol aggregation [32] or by substitution of DGEBA with TGDDM epoxy prepolymer [31]. As conclusion, the use of imidazolium ILMs as new monomers could be a promising route to achieve a nanoscale filler structuration in thermoset (epoxy) networks and to design nanomaterials combining properties from both ionic liquid and nanoclusters. Such nanomaterials could be useful as novel hybrid materials for electrochemistry applications.

### 3.2. Epoxy Conversion and Glass Transition Temperatures

It has been noticed that the presence of POSS^®^ cages on glass transition temperature of the hybrids is dependent on the nature of corner R groups on silsesquioxane cages and an epoxy matrix. The DSC thermograms of POSS^®Ph^-based hybrid O/I networks display a single glass transition temperature, i.e., the hybrid materials are homogenous [50]. The glass transition of the neat epoxy ILM/IPD networks occurred at 71 and at 54 °C for ILM1/IPD and ILM2/IPD networks, respectively (Appendix A). It can be noticed that all the hybrids O/I containing POSS^®Ph^-triol or IL-g-POSS^®Ph^ (whatever the chemical nature of the anion, chloride, or bis-triflimide-NTf_2_^−^ -) show a decrease of their glass transition temperature (60 and 45 °C for ILM1/IPD and ILM2/IPD networks, respectively) (Figure 2a,b). Nevertheless, the ILM2/IPD epoxy network containing POSS^®Ph^-triol displays higher T_g_ (74 °C) compared to the neat epoxy ILM2/IPD and the O/I hybrid containing IL-g-POSS^®Ph^ networks (Figure 2b).

It can be proposed that the decrease of the glass transition temperature results from the incomplete curing reaction of the ILMs monomers due to the presence of POSS^®Ph^ nanoclusters [62]. To confirm such an assumption, the epoxy conversion was determined by FTIR and the ^13^C HR-MAS NMR. As shown in Appendix A, a high conversion of epoxy-groups was obtained (>90%) and the total disappearance of the epoxy peaks (at 34, 47, and 52 ppm in the ^13^C-NMR spectra) after the curing procedure confirmed that the curing reactions are completed [46,47]. Therefore, the decrease of the glass transition temperatures cannot be associated to a non-fully completed curing reaction resulting from the incorporation of POSS^®Ph^-triol or IL-g-POSS^®Ph^ nanoclusters.

According to the literature, it has been reported for octa-aminophenyl POSS^®^ and octa-nitrophenyl POSS^®^-modified epoxies [50] that with a low content of POSS^®Ph^, a T_g_ decrease is observed. This decrease can be ascribed to the increase of free volume of the system due to the insertion of bulky POSS^®Ph^ cages between chains. According to the literature, two competitive factors can influence the glass transition temperatures. First, the steric hindrance effect of POSS^®^ cages on polymer chain motions will contribute to increase the glass transition temperature via reduction of local chain motions. Secondly, the inclusion of bulky POSS^®^ cages could give rise to the increase of free volume leading to a decrease of the T_g_. Nevertheless, the T_g_ depends on the nature of the interactions between POSS^®^ cages and polymer chains which are controlled by surrounding groups of the POSS^®^ cage, the nature of polymer matrix, and organic groups in POSS^®^ vertexes, etc. Thus, POSS^®^ could act on segmental motions of polymer chains as a nanofiller (surface interactions reducing mobility) or a plasticizer (increase in free volume). In fact, POSS^®Ph^-containing nanomaterials can display increased [50,63,64] or decreased [65,66] T_g_ compared to neat polymers. The introduction of POSS^®Ph^-triol or IL-g-POSS^®Ph^ in ILM1/IPD or ILM2/IPD networks led to an increase of free volume. The increased T_g_ for the O/I hybrid network containing POSS^®Ph^-triol based on ILM2/IPD matrix could be associated to the reactions between POSS^®Ph^-triol and epoxy groups.

Then, a decrease of the T_g_ could be attributed to: *i*/possible secondary reactions between IL-g-POSS^®Ph^ and epoxy ILM monomers generated by the anion (Cl^−^ or NTf_2_^−^) [54], or *iii*/secondary reactions between the hydroxyl silanol of the POSS^®Ph^ cages and the epoxy groups [63]. To confirm these assumptions, the remaining soluble fractions after curing schedule of the different systems were determined. In the case of epoxy ILM/IPD networks containing POSS^®Ph^-triol, the soluble fraction was determined to be 4.2% with ILM1/IPD and 4.0% with ILM2/IPD which means that some POSS^®Ph^-triol clusters (about 16 to 20%) are chemically attached to the epoxy networks. For the IL grafted-POSS^®Ph^, the soluble fraction was found for both epoxy ILM/IPD networks to be about 3.8 and 3.6% for IL.Cl-g-POSS^®Ph^ and for IL.NTf_2_-g-POSS^®Ph^, respectively. In addition, a covalent grafting of these IL-modified POSS^®Ph^ with the epoxy networks is ensured for about 24 to 28% of the nanoclusters, while the rest of POSS^®Ph^ remain non-bounded to the epoxy network. In summary, the incorporation of IL.Cl-g-POSS^®Ph^ and IL.NTf_2_-g-POSS^®Ph^ led to an increase of the free volume reducing the T_g_ of the networks.

### 3.3. Thermal Stability and Oxidative Resistance of Hybrid O/I Networks

Epoxies usually display a low flame retardancy. In addition, to obtain a higher thermostability of such polymer materials, multicomponent fire retardants are used, such as compounds containing Si, N, P, and halogens [67,68]. As described in our previous works [59], POSS^®Ph^ cages allow the generation of a ceramic-like surface layer during combustion which protects the inside matrix from oxidation limiting oxygen availability. In fact, such an inorganic surface barrier could limit the attack from the oxygen radical and the active radicals issued from degraded species, i.e., contributing to improve the resistance to oxidation. Moreover, the Cl^−^ and NTf_2_^−^ anions introduced via the ionic liquids can quench the active radical and limit the probability of chain reaction [69]. For such a behavior, the effects of IL-functionalized POSS^®^ on the thermal stability of networks based on ILM monomers have been investigated.

Thermal stability and thermo-oxidative resistance of the epoxy hybrid O/I networks were investigated from thermogravimetric analyses (TGA) under air (to favor oxidation reactions) and inert atmosphere (intrinsic degradation). According to such protocols, the influence of POSS^®Ph^-triol and IL-g-POSS^®Ph^ nanoclusters on the pyrolysis-oxidation resistance of the epoxy ILM/IPD networks was examined. This evaluation was undertaken from the weight change under air atmosphere as a function of the temperature, the initial decomposition temperature (T_d5%_), and the maximal degradation temperature (T_dmax_) (Figure 3a,b as well as Appendix A under inert atmosphere).

The thermal degradation (Appendix A) proceeds from two decomposition steps for the different epoxy hybrid O/I networks under nitrogen atmosphere. These steps occur between 250–400 °C and 400–550 °C defined from the vertex group degradation and dehydration-vaporization mechanism [70] at higher temperature (900 °C) producing a stable black residue (6 to 18 wt.%). Under air, a similar behavior is obtained. The O/I hybrid epoxy networks display the same decomposition profiles with a low degradation temperature (T_d5%_) similar to other POSS^®^-modified systems [71]. The thermograms also show a third decomposition step at higher temperature (from 550 to 750 °C) related to the maximum rate of vaporization and completed oxidation combustion reactions (Figure 3a,b) leading to almost no residue (from 0.3 to 2.3%). Residue production yield decreases for all O/I hybrid epoxy networks under air atmosphere compared to the ones obtained under nitrogen atmosphere. The ILM/IPD networks based on IL.NTf_2_-g-POSS^®Ph^ lead to more residue than the networks containing IL.Cl-g-POSS^®Ph^ or POSS^®Ph^-triol. The difference on the residue content could be explained by the competition between the sublimation and/or vaporization and the full oxidation reactions at high temperature in relation to the POSS^®Ph^ content, which are grafted to the epoxy networks. Thermal stability, degradation kinetics, and residue content of the O/I hybrid epoxy networks depend mainly on the nature of the organic groups surrounding the POSS^®Ph^ inorganic cage but also that to the chemical nature of the co-monomers considered to build the epoxy networks (ILM1 or ILM2).

As can be seen in Figure 3a,b the epoxy hybrid O/I networks containing POSS^®Ph^-triol or IL-g-POSS^®Ph^ nanoclusters with chloride or bistriflimide (NTf_2_^−^) as counter anions have a higher thermal stability (T_d5%_, T_dmax_) and a higher oxidation resistance than the neat epoxy networks (ILM1 or ILM2/IPD). Such behavior could be associated to the nanoscale dispersion of POSS^®Ph^-triol or IL-g-POSS^®Ph^ clusters in the epoxy matrices as well as to the improved interactions between the ILM monomers and POSS^®Ph^-POSS^®Ph^ aggregates. In addition, the increased T_d_ for epoxy hybrids containing POSS^®Ph^ could result from the increased chain spacing which modify also the thermal conductivity of POSS^®Ph^-modified networks. The POSS^®^ ceramization is another important effect which could contribute to improving the thermostability. In general, POSS^®^ cage as a ceramic [72] precursor forms silica layer after completed oxidation. After the emission of organic groups in POSS^®Ph^ cages, the silica layer seals the polymer, retards the oxidation of bulk polymer. On the other hand, POSS^®^ is a surface-active moiety, tending to migrate to the surface as a tenside [73]. It is worth of underlining that the O/I hybrid epoxy networks based on ILM2 exhibit the highest thermal stability (T_d5%_) and resistance towards oxidation (DTG peak concerning the network contained IL.NTf_2_-g-POSS^®Ph^, 665 °C with ILM2/IPD against 573 °C with ILM1/IPD matrix) compared to the epoxy hybrid O/I networks based on ILM1.

### 3.4. Surface Properties of Hybrid O/I Networks

Total surface energies of the epoxy hybrid O/I networks containing POSS^®Ph^-triol, IL.Cl-g-POSS^®Ph^, and IL.NTf_2_-g-POSS^®Ph^ prepared from co-polymerization of ILM1 or ILM2 with IPD monomers are summarized in Table 2.

First, the results revealed that the substitution of DGEBA with the ILMs and the introduction of 5 wt.% of POSS^®Ph^-triol or IL-g-POSS^®Ph^ (with Cl^−^ or NTf_2_^−^ anion) considerably increased the contact angles formed by the probe liquids at the surface of the epoxy networks (ILM/IPD). Taking into account the increase of the contact angles of these two probe liquids on ILM/IPD/POSS^®Ph^ surfaces, one can conclude that the introduction of the POSS^®Ph^ and the substitution of DGEBA by ILM monomers increase the hydrophobic and oleophobic characteristics of the surface. In addition, it is interesting to observe that the values obtained are similar to ones measured for different fluorinated polymers such as PTFE, i.e., 110–120° for water and 82–83° for methylene diiodide [74,75] confirming the hydrophobic character of the epoxy network surfaces.

A more detailed analysis taking into account the polar components and dispersive of the surface energy of the epoxy hybrid O/I networks shows that the dispersive component is the major contribution of the surface energy (98%). Consequently, the interactions initiated by this type of surface are mainly London dispersive interactions. As DGEBA monomer is replaced by ILMs and modified by the incorporation of the POSS^®Ph^ nanoclusters, a significant decrease of the dispersive component is observed (from 35 mJ.m^−2^ for DGEBA/IPD (see Table 2) to 18 mJ.m^−2^ for ILM2/IPD/IL.NTf_2_-g-POSS^®Ph^). At the same time, the non-dispersive component of the initially low surface energy is also decreased. The London dispersive interactions are still predominant but are reduced by the addition of the POSS^®Ph^ leading to a decrease of the total surface energy. These results can be explained by the hydrophobic contribution brought by the POSS^®Ph^ molecules having phenyl ligands [76] and epoxy ionic liquids monomers containing imidazolium units with NTf_2_^−^ anions [77,78]. These units are well-known to lower surface tension [79]. In addition, the ILM/IPD networks containing POSS^®Ph^-triol or IL-g-POSS^®Ph^ networks display a more pronounced hydrophobic character than one of the DGEBA/IPD modified by POSS^®Ph^ network. These new results combined with the high thermal stability open new and promising routes in the preparation of surface coatings with self-cleaning and anti-corrosion properties for automotive, aerospace, or electronics applications.

### 3.5. Ionic Conductivity of Hybrid O/I Networks

In order to characterize the ILM/IPD epoxy networks and the corresponding hybrid networks for potential solid polymer electrolytes applications, dielectric measurements under AC voltage were carried out. Thus, the influence of IL-g-POSS^®Ph^ nanoclusters could be analyzed. The real, *ε*′_r_, and the imaginary, *ε*″_r_, parts of the complex relative permittivity were extracted from measurements undertaken under 5 V sinusoidal applied voltage (Figure 4a,b). In order to evaluate the DC conductivity (quasi-steady state) at high temperatures, it is required to start by evaluating the changes taking place in the AC conductivity domain. Figure 5 gives the changes of AC conductivity as function of frequency for the epoxy DGEBA and ILM1 or ILM2 cured with IPD as well as the different hybrid O/I networks obtained from the addition of IL-g-POSS^®Ph^ with the two different anions: chloride (Cl^−^) or bis-trifluoromethanesulfonimidate (NTf_2_^−^). Taking into account the fact that the glass transition temperatures of the ILM/IPD networks and their corresponding O/I hybrid networks, which vary from 45 to 71 °C (see Figure 3 and Appendix A), the conductivity was measured from −80 up to 200 °C for the DGEBA/IPD and up to 150 °C for ILM/IPD (the same for their hybrid O/I networks).

The neat epoxy network (DGEBA/IPD) shows relative permittivity, *ε*′_r_, starting from 3.5 at low temperature and up to 6 at temperatures higher than room temperature. A slight increase with a maximum value close to 170 °C corresponds to the α relaxation of the epoxy system. The ILM/IPD and their corresponding O/I hybrids obtained by the introduction of 5 wt.% of IL-g-POSS^®Ph^ nanoclusters with the chloride or bistriflimide (NTf_2_^−^) anions have relative permittivity, *ε*′_r_, much higher (from 30 to 50%) at room temperature and also show an exponential increase after the glass transition temperature (between 50 and 80 °C depending on the epoxy system) to achieve high values at higher temperatures. The shape of the curves is not influenced by the addition of IL-g-POSS^®Ph^ but only shifted to higher values (Figure 4). The imaginary part of the relative permittivity values, *ε*_r_”, shows a similar behavior at room temperature and above glass transition temperature. Figure 4b shows just one broad transition with peaks located close to −50 °C corresponding to β relaxation associated to the motion of the hydroxyethyl units generated from the crosslinking reaction [80]. The starting of the increase from 30 to 50 °C, depending on the system, is generally due to the relaxation processes of the main chains of the crosslinked epoxy network (α relaxation). This relaxation is not well visible on the plots (Figure 4b) due to the important conduction phenomenon that gives an exponential variation towards higher values at the same temperature range. This relaxation arises from the fact that free charges are immobilized in the network and at sufficiently high temperatures, i.e., above T_g_ of the polymer, the charges can migrate in the presence of an applied electric field. As the temperature increases above the glass transition temperature, the conductivity of the system increases giving rise to the increase of the concentration of charge carriers [81]. This increase of the dielectric constant can be explained by (*i*) the nanoscale dispersion of IL-g-POSS^®Ph^ nanoclusters inside the epoxy matrix leading to limited free volume, and by (*ii*) the increased polarizability of the resulting hybrid nanomaterials since they have the highest ionic liquid content (cation/anion). This allows the formation of conductive interfacial layers around aggregated POSS^®Ph^, i.e., a potential conductive path. Hence, the development of ILM/IPD/IL-g-POSS^®Ph^ nanomaterials with high dielectric constant is expected to find application as suitable electrolyte material.

The σ_AC_ conductivity of the networks showed a linear increase with respect to frequency with a dependence equal to one at high frequencies. Frequency dependency follows the law (2):σ_AC_ (ω) = ω^s^(2)
with 0 ≤ s ≤ 1 characterizing hopping conduction [82,83]. As the temperature increases, the conductivity becomes independent on the frequency and increases by 2 to 3 orders of magnitude for all the epoxy networks. A horizontal plateau appears progressively, expressing the thermal activation of the DC conductivity. The amplitude of this plateau corresponds to the DC conductivity at a given temperature. As the temperature increases, the mobility of the electrons increases shifting the plateau to higher frequencies and higher value of conductivity. The plateau appearing for different epoxy ILM/IPD and the corresponding hybrid O/I networks seem to have the same influence with respect to frequency and temperature. A decrease of the conductivity values is observed at high temperatures and low frequencies, such as for ILM1/IPD, ILM2/IPD, and ILM2/IL.NTf_2_-g-POSS^®Ph^ over 120 °C and 10^2^ Hz. This can be attributed to the electrode polarization phenomenon leading to blocking electrodes [84]. Conductivity DC values are extrapolated before this decrease. As a result, the ionic conductivity for all investigated epoxy ionic liquids networks from the plateau region of each dielectric spectrum at room or high temperature follow the following order: ILM2/IPD/IL.NTf_2_-g-POSS^®Ph^ > ILM2/IPD/IL.Cl-g-POSS^®Ph^ > ILM1/IPD/IL.NTf_2_-g-POSS^®Ph^ > ILM1/IPD/IL.Cl-g-POSS^®Ph^ > ILM2/IPD > ILM1/IPD > DGEBA/IPD, which is the reverse order of T_g_ shown in Figure 3. These results suggest that polymer chain relaxation plays a dominant role in ion transportation. Specifically, the ILM/IPD/IL-g-POSS^®Ph^ with NTf_2_^−^ anions networks show significantly a higher ionic conductivity compared to the ILM/IPD/IL-g-POSS^®Ph^ with Cl^−^ anion networks, indicating also a strong dependence of the conductivity on the identity of the counterion which coincides with the significantly lower T_g_. This can be attributed to the difference in symmetry and size of the anions and to the dissociation energy of the ion pairs when the effects of T_g_ are removed. Chen et al. [85] observed a similar effect on ionic conductivity in a PIL when they exchanged chloride (Cl^−^) anion with the bis(trifluoromethanesulfonyl)imide (NTf_2_^−^) anion. In addition, with increasing temperature, the difference of ionic conductivity between the ILM/IPD and their hybrid O/I networks with different anions decreases, suggesting that at high temperatures the importance of T_g_ becomes less pronounced. Therefore, ionic conductivity values are included on a range extending from 3.40 × 10^−8^ up to 9.37 × 10^−5^ S.m^−1^ at 40 °C as a function of the system (Table 3). Moreover, for high temperatures above the T_g_, ionic conductivity increased significantly to reach much higher values (see Table 3 and Figure 5).

Figure 6 presents the evolution of the DC conductivity of all the epoxy ionic liquid networks as a function of temperature extrapolated from dielectric spectroscopy measurements at 0.1 Hz. The measured DC conductivity by dielectric spectroscopy has a linear slope above the glass transition temperature of epoxy for DGEBA/IPD (Appendix A), ILM/IPD and their epoxy hybrid O/I networks (Figure 6). All the slopes are well fitted with the Arrhenius (Equation (3)) law by exhibiting linear dependence, where the obtained Arrhenius 2-parameters, i.e., activation energies and infinite conductivity, σ_0_, are presented in Table 4 calculated from the slope of the fitted curves in Figure 6.

(3)$$\sigma_{DC}(T)\;=\;\sigma_{0}\;\exp\;(-\frac{E_{a}}{\;k_{B}T})$$
where *k_B_* is the Boltzmann’s constant (*k_B_* = 8.617 × 10^−5^ eV. K^−1^).

The R-square values confirm that the temperature dependence of ionic conductivity is very good in accordance with the Arrhenius equation. Conductivity increases with increasing temperature and follows Arrhenius dependence for all nanomaterials, resulting in an accelerated motion of polymer chains and promoting faster migration of ions at higher temperatures [86,87]. Thus, the mechanism of ionic conductivity can be understood from activation energy (*E_a_*).

A lower activation energy, E_a__,_ means a better transmission conductor for DGEBA/IPD, ILM/IPD, and their hybrid O/I networks presented in Table 4. The activation energies are in the same range for the ILM1/IPD and ILM2/IPD networks and decrease for both with the addition of IL-g-POSS^®Ph^ whatever the anion nature. This signifies that the substitution of ILM1 with ILM2 affected neither the DC conductivity values nor the activation energy of the system. The lowest value of E_a_ is for ILM/IPD/IL-g-POSS^®Ph^ networks as is observed in Table 4, and the highest ionic conductivities are also observed for the same networks as presented in Table 3. Compared to pure ILs, the activation energies for ionic conductivities with IL-g-POSS^®Ph^ nanoclusters (Table 4) are higher [86,87]. In fact, this behavior indicated that the introduction of IL-g-POSS^®Ph^ led to an increase of ion transportation due to the structural changes (clustering) in the network. This is due to decrease of T_g_ of the ILM/IPD networks by addition of IL-g-POSS^®Ph^ and the presence of high amount of ionic liquid facilitating ionic movement in the matrix. As a result, all investigated epoxy ionic liquids materials displayed a very good ion transmission performance with the maximum ionic conductivity in the temperature range, from 40 to 80 °C.

## 4. Conclusions

In this work, a new generation of epoxy hybrid organic/inorganic networks based on diepoxydized ionic liquid monomers (ILMs) and POSS^®Ph^ nanoclusters,—i.e., a non-fully condensed polyhedral oligomeric silsesquioxane (POSS^®Ph^-triol) or IL-modified POSS^®Ph^ combined with two different anions chloride (Cl^−^) or bis-trifluoromethanesulfonimidate (NTf_2_^−^)—has been prepared. Either POSS^®Ph^-triol or IL-g-POSS^®Ph^ could generate crosslinks from a pre-reaction with the epoxy prepolymers, ILM1 or ILM2. The analysis of morphologies of hybrid nanomaterials was performed by transmission electron microscopy which evidenced well-dispersed POSS^®^-aggregates-based nanostructured networks with nanometer scale particle size of POSS^®Ph^ aggregates in the epoxy ILM/IPD matrix. As a consequence, the nanostructured epoxy O/I hybrids display a decrease of the glass transition as compared to the neat ILM/IPD networks. Nevertheless, changing the chemical architecture of ILMs monomers, the introduction of POSS^®Ph^-triol (ILM2/IPD) permits balancing the T_g_ decrease (74 vs. 54 °C). The epoxy hybrid O/I networks display higher thermal stability and resistance to oxidation, especially the ILM2/IPD matrix. Moreover, these nanostructured materials display low surface energy (18.2–20.7 mJ.m^−2^), high hydrophobicity (H_2_O contact angle from 108 to 120°) and oleopophobicity (methylene diiodide contact angle from 75 to 92°) which make them promising materials for coating applications. Finally, the epoxy-LI (ILMs/IPD) networks loaded with IL-g-POSS^®Ph^ appear to be promising materials for reducing charge accumulation within insulation. They have the advantage of reaching interesting conductivities for low glass transition temperatures (around 70 °C). The contribution is perceptible from temperatures well below the glass transition, with conductivity at 40 °C of the order of 9.37 × 10^−5^ S.m^−1^ against 6.84 × 10^−2^ S.m^−1^ at 100 °C. Although the charge carriers are not clearly identified, the improvement seems essentially of ionic origin. On the other hand, a thermal activation mechanism has been demonstrated. These results are promising for new solid polymer electrolytes.

## Figures and Tables

**Figure 1 nanomaterials-12-00550-f001:**
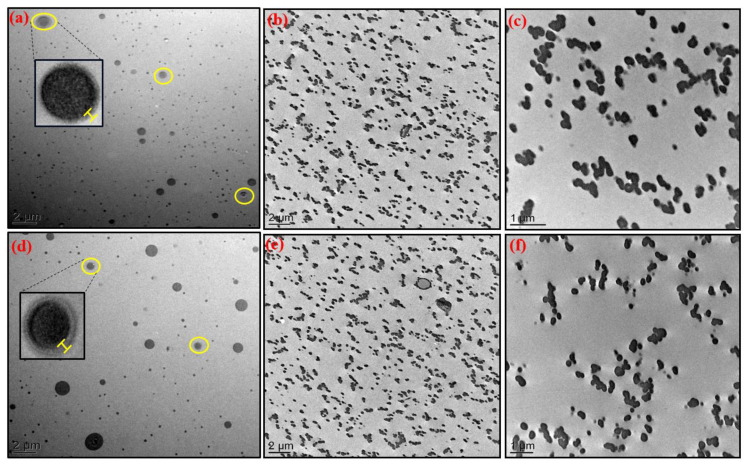
TEM images of the different epoxy ILM hybrid O/I networks containing POSS^®Ph^-triol, IL.Cl-g-POSS^®Ph^ and IL.NTf_2_-g-POSS^®Ph^, respectively, prepared with epoxy monomer ILM1 (**a**–**c**), and with ILM2 (**d**–**f**).

**Figure 2 nanomaterials-12-00550-f002:**
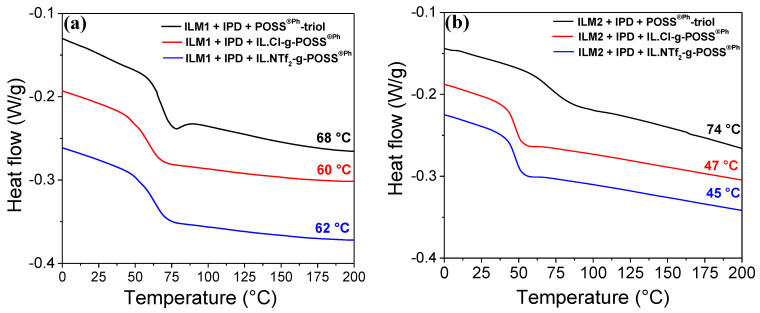
DSC traces of the epoxy hybrid O/I networks based on epoxy IL monomers (**a**) ILM1, and (**b**) ILM2–containing POSS^®Ph^-triol, IL.Cl-g-POSS^®Ph^, and IL.NTf_2_-g-POSS^®Ph^, respectively.

**Figure 3 nanomaterials-12-00550-f003:**
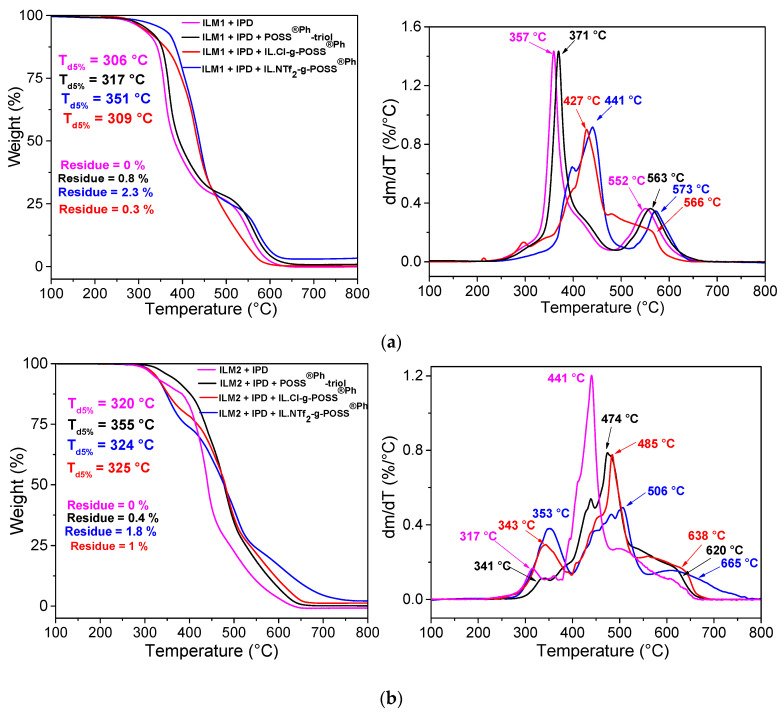
Evolution of weight loss (TGA) and derivative of TGA curves (DTG) as a function of the temperature for the epoxy hybrid O/I networks (prepared based on epoxy monomers: (**a**) ILM1, and (**b**) ILM2) containing POSS^®Ph^-triol, IL.Cl-g-POSS^®Ph^, and IL.NTf_2_-g-POSS^®Ph^, respectively.

**Figure 4 nanomaterials-12-00550-f004:**
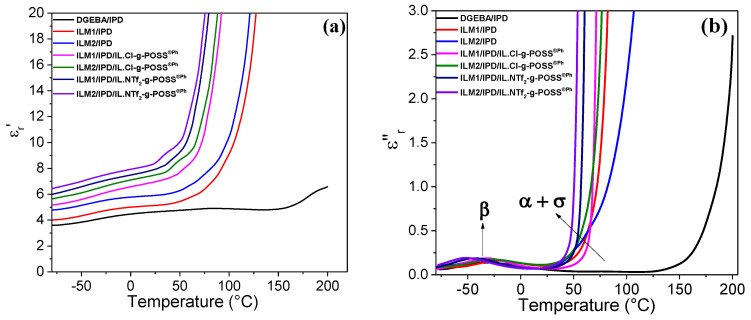
Evolution of relative real permittivity (**a**), and imaginary part (**b**) as a function of temperature at 100 Hz of DGEBA/IPD network, ILM/IPD network, and the corresponding hybrid O/I epoxy networks.

**Figure 5 nanomaterials-12-00550-f005:**
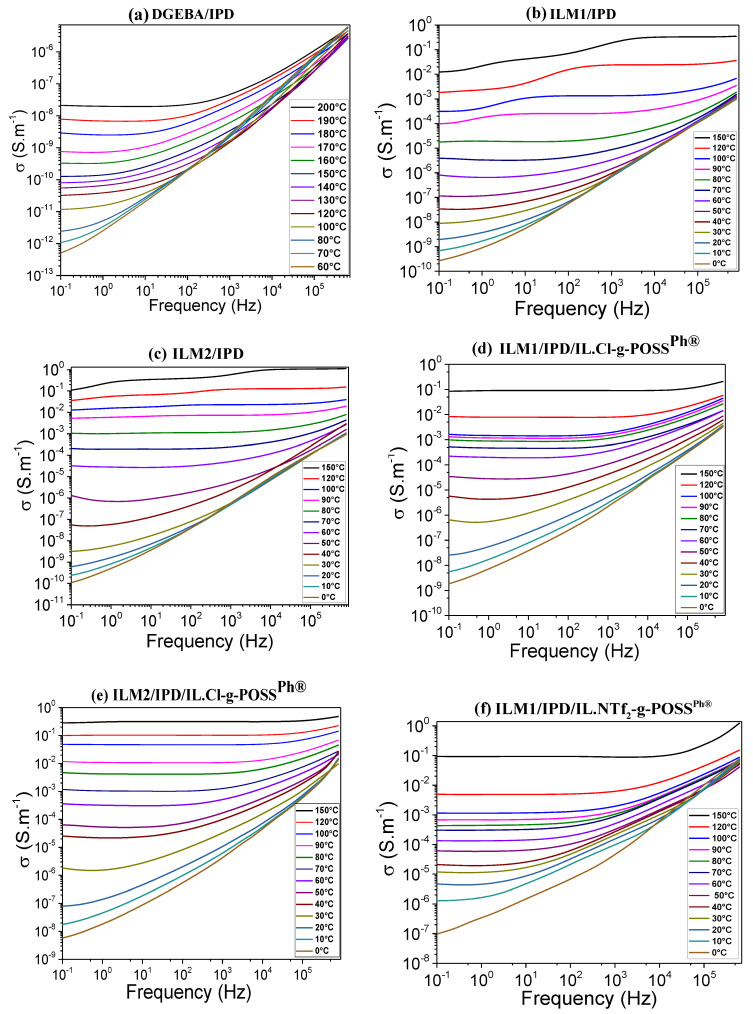
Evolution of AC conductivity as a function of frequency for (**a**) DGEBA/IPD, (**b**) ILM1/IPD, (**c**) ILM2/IPD, (**d**) ILM1/IPD/IL.Cl-g-POSS^®Ph^, (**e**) ILM2/IPD/IL.Cl-g-POSS^®Ph^, (**f**) ILM1/IPD/IL.NTf_2_-g-POSS^®Ph^, and (**g**) ILM2/IPD/IL.NTf_2_-g-POSS^®Ph^.

**Figure 6 nanomaterials-12-00550-f006:**
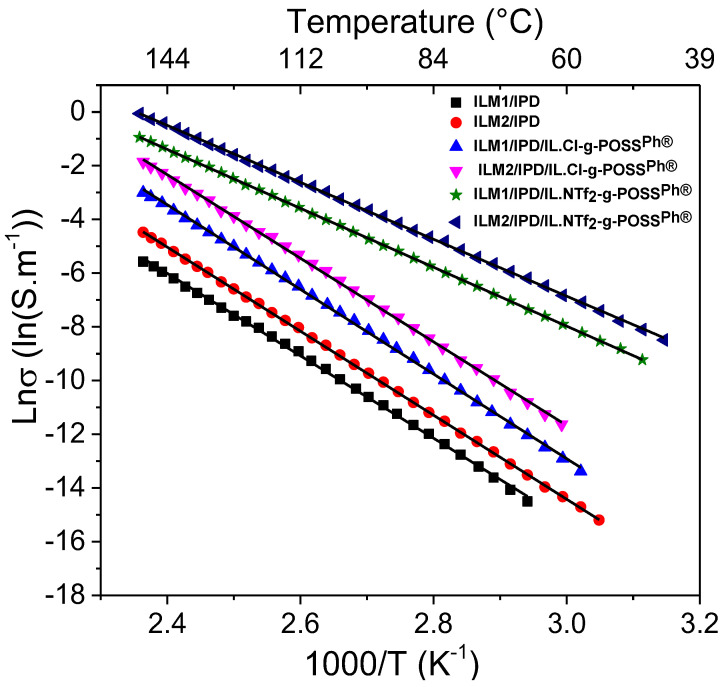
Dependence of DC conductivity with temperature (extrapolated from AC conductivity values at 0.1 Hz). Solid black lines represent a regression to the Arrhenius equation.

**Table 1 nanomaterials-12-00550-t001:** Chemical formula of components used to design nanostructured epoxy-amine networks.

Name	Chemical Formula	Characteristics
Diglycidyl ether ofbisphenol A(**DGEBA**)	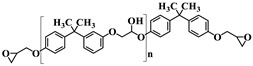 n = 0.15	377 g.mol^−1^,eew: 185–192 g/eq ^a^,T_g_ = −13 °C ^b^,T_d_ = 330 °C ^b^
3-[2-(oxiran-2-yl)ethyl]-1-[6,7]imidazolium 1,1,1-trifluoroN-[(trifluoromethyl)sulfonyl]methanesulfonamide**ILM1**	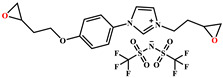	Prepared following published procedure [28]T_d_ = 489 °C ^b^
3,3′-(1,4-butanediyl)bis [1-(4-(2-(oxiran-2-yl)ethyl)phenyl)]imidazolium 1,1,1-trifluoro-*N*-[(trifluoromethyl)sulfonyl]methanesulfonamide**ILM2**	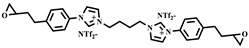	Prepared following published procedure [46]T_m_ = 71 °C ^a^T_d_ = 460 °C ^b^
Isophoronediamine(**IPD**)	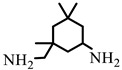	M = 170.3 g.mol^−1^
Heptaphenyl-trisilanol POSS^®^ (**POSS^Ph^-triol**)	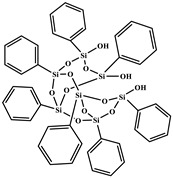	M = 931.34 g mol^−1^,T_m1_ = 217 °C ^a^,T_m2_ = 230 °C ^a^,T_d_ = 632 °C ^b^
1-methyl-3-propyl heptaphenyl octasilesquioxane imidazolium chloride(**IL.Cl-g-POSS^®Ph^**)	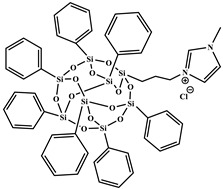	Prepared following the procedure reported in [45]M = 1116.04 g mol^−1^,T_m1_ = 127 °C ^a^,T_m2_ = 168 °C ^a^,T_d_ = 632 °C ^b^
1-methyl-3-propyl heptaphenyloctasilesquioxane imidazoliumbis(trifluoromethane)sulfonimide(**IL.NTf_2_-g-POSS^®Ph^**)	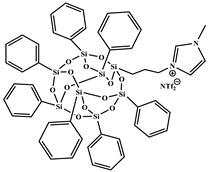	Prepared following the procedure reported in [45]M = 1360.73 g mol^−1^,T_m1_ = 135 °C ^a^T_m2_ = 158 °C ^a^,T_d_ = 534 °C ^b^

^a^ T_m_: Melting temperature, ^b^ T_d_: Degradation temperature determined at the maximum of the first derivative of the weight loss as a function of temperature.

**Table 2 nanomaterials-12-00550-t002:** Determination of dispersive and non-dispersive components of the surface energy on the neat epoxy DGEBA/IPD, ILM/IPD and the corresponding hybrid O/I networks from contact angle with water and methylene diiodide at room temperature.

Samples	Θ_Water_ (°)	Θ_CH2I2_ (°)	γ_non-dispersive_ (mJ.m^−2^)	γ_dispersive_ (mJ.m^−2^)	γ_total_ (mJ.m^−2^)
DGEBA/IPD	79	49	4.8	34.8	39.6
ILM1/IPD	102	62	0.8	24.6	25.4
ILM1-IPD/POSS^®Ph^-triol	108	75	0.3	20.4	20.7
ILM1-IPD/IL.Cl-g-POSS^®Ph^	110	78	0.3	19.9	20.2
ILM1-IPD/IL.NTf_2_-g-POSS^®Ph^	112	82	0.2	19.6	19.8
ILM2/IPD	106	72	0.6	21.0	21.6
ILM2-IPD/POSS^®Ph^-triol	115	86	0.2	19.4	19.6
ILM2-IPD/IL.Cl-g-POSS^®Ph^	116	88	0.2	18.8	19.0
ILM2-IPD/IL.NTf_2_-g-POSS^®Ph^	120	92	0.2	18.0	18.2

**Table 3 nanomaterials-12-00550-t003:** Ionic conductivity of the different epoxy ionic liquids networks at different temperatures.

Samples	σ_AC_ @ 40 °C	σ_AC_ @ 50 °C	σ_AC_ @ 70 °C	σ_AC_ @ 100 °C
ILM1-IPD	3.40 × 10^−8^	1.15 × 10^−7^	3.91 × 10^−6^	3.12 × 10^−4^
ILM1-IPD/IL.Cl-g-POSS^®Ph^	5.74 × 10^−6^	3.46 × 10^−5^	3.05 × 10^−4^	1.63 × 10^−2^
ILM1-IPD/IL.NTf_2_-g-POSS^®Ph^	2.11 × 10^−5^	6.12 × 10^−5^	6.28 × 10^−4^	2.15 × 10^−2^
ILM2-IPD	5.42 × 10^−8^	3.80 × 10^−6^	2.08 × 10^−4^	1.28 × 10^−3^
ILM2-IPD/IL.Cl-g-POSS^®Ph^	2.54 × 10^−5^	8.39 × 10^−5^	1.19 × 10^−3^	4.38 × 10^−2^
ILM2-IPD/IL.NTf_2_-g-POSS^®Ph^	9.37 × 10^−5^	9.81 × 10^−4^	7.27 × 10^−3^	6.84 × 10^−2^

**Table 4 nanomaterials-12-00550-t004:** Arrhenius equation regression values of temperature dependent conductivity data at room temperature.

Samples	σ_0_ (S.m^−1^)	E_a_ (eV)	R-Square
ILM1-IPD	4.87 × 10^13^	1.36	0.99856
ILM1-IPD/IL.Cl-g-POSS^®Ph^	2.71 × 10^15^	1.14	0.99612
ILM1-IPD/IL.NTf_2_-g-POSS^®Ph^	1.14 × 10^15^	1.06	0.99712
ILM2-IPD	6.15 × 10^13^	1.35	0.99875
ILM2-IPD/IL.Cl-g-POSS^®Ph^	7.72 × 10^10^	0.92	0.99436
ILM2-IPD/IL.NTf_2_-g-POSS^®Ph^	3.28 × 10^10^	0.86	0.99236

## Data Availability

The data presented in this study are available on request from the corresponding author.

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
