# Peer review of "New Epoxy Thermosets Organic-Inorganic Hybrid Nanomaterials Derived from Imidazolium Ionic Liquid Monomers and POSS®Ph"

_nanomaterials, 2022, doi:10.3390/nano12030550_

Round 1
Reviewer 1 Report
In this manuscript Chabane et al. have combined POSS with epoxy IL monomers to prepare nanostructured epoxy PILs with interesting properties. The manuscript is well written, is understandable, and brings new knowledge into the field. I recommend publication of this manuscript subject to the following revisions:
- Page 6, the interface thickness is reported to be 10-30 nm. (a) Is this the result of direct experimental measurements? If yes, please include more details in the text. (b) The interface thickness for polymers at the interface of a solid substrate is reported to vary from a few nanometers to 2-3 times the radius of gyration of the chain, depending on the property under investigation (see for example Phys. Chem. C2013, 117, 5249). Based on this, I assume that an interface thickness of 30 nm for monomers of ILs at the interface of nanoclusters, studied in this work, in a bit large. It would be nice if the authors could comment on this point in the text.
- Pages 7 and 8, the decrease in Tg of the epoxy network containing POSS is partly attributed to the incomplete phase separation. Considering the fact that even at high temperatures some signatures of clustering (nanoscale phase separation) is seen, decreasing the temperature toward the Tg reduces the barrier height for phase separation considerably, as the barrier height strongly depends on the temperature (see for example, Chem. Theory Comput.2019, 15, 1345). This means that if the phase separation is going to be an important issue in the present work, lowering the temperature leads to the formation of a completely phase separated system. In other words, I think in this work some sort of clustering (shown in the TEM images) occurs and the system does not undergo phase separation in the temperature window studied. I mostly resonate with the increased free volume effect (which decreases the Tg) and is also in line with the increased hydrophobicity, as discussed by the authors.
- One of the interesting features of the epoxy PILs, synthesized in this work, is enhancement of the ionic conductivity, compared to monomers of ILs. (a) As the motion of ions (cations and anions) at the interface is restricted (compared to the neat ILs), how do the authors justify the increase in the ionic conductivity? (b) the activation energies for ionic conductivities (Table 4) are much larger than previous reports in the literature for pure ILs (J. Am. Chem. Soc. 2009, 131, 15825). Much larger activation energies, compared to pure ILs, could be due to the structural changes (clustering) in the epoxy PIL systems studied in this work. If possible, please comment on this point in the manuscript.
Author Response
In this manuscript Chabane et al. have combined POSS with epoxy IL monomers to prepare nanostructured epoxy PILs with interesting properties. The manuscript is well written, is understandable, and brings new knowledge into the field. I recommend publication of this manuscript subject to the following revisions:
1- Page 6, the interface thickness is reported to be 10-30 nm. (a) Is this the result of direct experimental measurements? If yes, please include more details in the text. (b) The interface thickness for polymers at the interface of a solid substrate is reported to vary from a few nanometers to 2-3 times the radius of gyration of the chain, depending on the property under investigation (see for example Phys. Chem. C2013, 117, 5249). Based on this, I assume that an interface thickness of 30 nm for monomers of ILs at the interface of nanoclusters, studied in this work, in a bit large. It would be nice if the authors could comment on this point in the text.
Thank you very much for your comment. Concerning your first question, we have determined one estimation of the interface and not the interphase by Image J Software from the TEM micrographs. For b), We agree with you and for this reason, we have added more details in the experimental measurements. The interface that we estimated between 10 and 30 nm does not correspond to the interface of a nanocluster but at 2% from the total volume of the POSS®Ph-triol aggregate. This can be explained by the low fraction of POSS®Ph-triol nanoclusters reacted with the epoxy-amine matrix.
2- Pages 7 and 8, the decrease in Tg of the epoxy network containing POSS is partly attributed to the incomplete phase separation. Considering the fact that even at high temperatures some signatures of clustering (nanoscale phase separation) is seen, decreasing the temperature toward the Tg reduces the barrier height for phase separation considerably, as the barrier height strongly depends on the temperature (see for example, Chem. Theory Comput.2019, 15, 1345). This means that if the phase separation is going to be an important issue in the present work, lowering the temperature leads to the formation of a completely phase separated system. In other words, I think in this work some sort of clustering (shown in the TEM images) occurs and the system does not undergo phase separation in the temperature window studied. I mostly resonate with the increased free volume effect (which decreases the Tg) and is also in line with the increased hydrophobicity, as discussed by the authors.
Thank you very much for your comment and yes, we agree with you. According to the literature on epoxy-POSS nanocomposites, the different authors have explained also the decrease of Tg by an increase of the free volume. We have removed our assumptions to the incomplete phase separation.
3- One of the interesting features of the epoxy PILs, synthesized in this work, is enhancement of the ionic conductivity, compared to monomers of ILs. (a) As the motion of ions (cations and anions) at the interface is restricted (compared to the neat ILs), how do the authors justify the increase in the ionic conductivity? (b) The activation energies for ionic conductivities (Table 4) are much larger than previous reports in the literature for pure ILs (J. Am. Chem. Soc. 2009, 131, 15825). Much larger activation energies, compared to pure ILs, could be due to the structural changes (clustering) in the epoxy PIL systems studied in this work. If possible, please comment on this point in the manuscript.
- a) Yes, we agree with you, we believe that the increase in the ionic conductivity values was due to the free ionic liquid due to the different counter anions presented on POSS nanoclusters and their nanostructuring inside the matrix which favors the creation of an ionic conduction pathway. b) Regarding the activation energies values of the resulted hybrid epoxy networks, we have added more explanations in the manuscript to better show the importance of these new results.
Reviewer 2 Report
The manuscript's scope is important, particularly in the light of the growing interest in solid electrolytes as fundamental components in electrochemical devices such as lithium-ion batteries, fuel cells, solar cells because of their key role in determining devices performance, durability and safety.
The whole idea behind the manuscript is very interesting but, the manuscript as a whole needs to be revised making the text easier to read.
Extensive editing of the English language and style is required throughout the manuscript
As the authors did NOT insert line numbers throughout the manuscript (which is very inconvenient for commenting), relevant parts of text to be commented are marked by page numbers in parentheses (e.g. "(P. 9)") or by the section title (e.g., "(TITLE)").
“abstract” needs to be revised to lets the readers get the essence of the paper quickly. The very first part should contextualize the research challenge while the rest of the abstract should summarize the key points from the paper and the conclusions.
“introduction “The introduction focuses mainly on papers dated mid-2010s, probably the analysis of recent publications will improve the paper outcome description.
“2.3. Determination of the soluble fraction extractable from hybrid O/I networks”
The method description is lacking details (i.e. how the THF was removed and if the method were replicated or not, the precision and accuracy of the measures is crucial in this kind of experiment and must be highlighted) “extractable” is not the appropriate term, please revise the title.
“FTIR” was mentioned but no spectra or discussions are present.
The description of the methods and the experimental design are not exhaustive and clearly described. The whole presentation of the characterization experiments needs an improvement.
Both in “abstract” and “conclusion” the following sentence “In this work, various epoxy hybrid organic/inorganic nanomaterials based on ILMs were synthesized considering 5 wt. % of non-fully condensed polyhedral oligomeric silsesquioxane (POSS®Ph-triol) or IL-modified POSS®Ph combined with two different anions chloride (Cl-) or bis-trifluoromethanesulfonimidate (NTf2-). POSS®Ph-triol or IL-gPOSS®Ph could generate crosslinks from a pre-reaction with the epoxy prepolymers, ILM1 or ILM2.” should be redraft because “various” is too generic and the meaning is not clear.
In my opinion, the manuscript may be reconsidered after major revision for a second round.
Author Response
The manuscript's scope is important, particularly in the light of the growing interest in solid electrolytes as fundamental components in electrochemical devices such as lithium-ion batteries, fuel cells, solar cells because of their key role in determining devices performance, durability and safety.
1-“abstract” needs to be revised to lets the readers get the essence of the paper quickly. The very first part should contextualize the research challenge while the rest of the abstract should summarize the key points from the paper and the conclusions.
Thank you very much for your comment, the abstract has been changed in consequence.
2-“introduction “The introduction focuses mainly on papers dated mid-2010s, probably the analysis of recent publications will improve the paper outcome description.
Thank you very much for your comment, we have added some more recent papers to improve the paper outcome description, i.e. to develop solid electrolytes (one paragraph). But, this paper is also based on the POSS explaining that the papers dated of mid-2010s.
3- The method description is lacking details (i.e. how the THF was removed and if the method were replicated or not, the precision and accuracy of the measures is crucial in this kind of experiment and must be highlighted) “extractable” is not the appropriate term, please revise the title.
Thank you very much for your comment, we have added more details about the method in the manuscript. The THF was removed by evaporation using a rotary evaporator; then the obtained fraction was also dried at 60° C for 48 h and analyzed by FTIR and C-NMR. This method is well-known and has already been widely reported in the literature.
4-“FTIR” was mentioned but no spectra or discussions are present.
In fact, FTIR was mentioned in the discussion and the FTIR spectra of the epoxy hybrid O/I networks are presented in the supporting information. To confirm such an assumption, the epoxy conversion was determined by FTIR and the 13C HR-MAS NMR. As shown in Figures S2 and S3 (see supporting information), a high conversion of epoxy-groups was obtained (> 90 %) and the total disappearance of the epoxy peaks (at 34, 47, and 52 ppm in the 13C-NMR spectra) after the curing procedure confirmed that the curing reactions are completed [46, 47].
5- The description of the methods and the experimental design are not exhaustive and clearly described. The whole presentation of the characterization experiments needs an improvement.
6- Both in “abstract” and “conclusion” the following sentence “In this work, various epoxy hybrid organic/inorganic nanomaterials based on ILMs were synthesized considering 5 wt. % of non-fully condensed polyhedral oligomeric silsesquioxane (POSS®Ph-triol) or IL-modified POSS®Ph combined with two different anions chloride (Cl-) or bis-trifluoromethanesulfonimidate (NTf2-). POSS®Ph-triol or IL-gPOSS®Ph could generate crosslinks from a pre-reaction with the epoxy prepolymers, ILM1 or ILM2.” should be redraft because “various” is too generic and the meaning is not clear.
Thank you very much for all your suggestions, all the co-authors have fully revised the manuscript and we have tried to correct as many sentences as possible. Thank you for your comprehension.
Reviewer 3 Report
In this work, various epoxy hybrid organic/inorganic nanomaterials based on ILMs were synthesized containing 5 wt% of non-fully condensed polyhedral oligomeric silsesquioxane (POSS®Ph-triol) or IL-modified POSS®Ph combined with two different anions chloride (Cl-) or bis-trifluoromethanesulfonimidate (NTf2-). The morphologies and performances these hybrid nanomaterials were characterized and analyzed. Both the epoxy matrix and POSS filler contained ion liquid unit, which is different from the previous epoxy system. Overall, the work is interesting and well organized. It could be considered for acceptance after a minor revision.
- In Table 1, the value of M = 1.116.04 g mol-1 and M = 1.360.73 g mol-1 contained unnecessary dot. Please check the molecular weight and correct it.
- In Table 1, the chemical formula of POSSPh-triol, IL.Cl-g-POSS®Ph and IL.NTf2-g-POSS®Ph needed to be presented in the same font as ILM molecules.
- In Figure 1, it is better to show Figure 1c and 1f using a same scale bar as Figure 1a and 1b.
- The dielectric constant of the hybrid materials should be added for a better evaluation for their possible application in the electrochemistry.
Author Response
In this work, various epoxy hybrid organic/inorganic nanomaterials based on ILMs were synthesized containing 5 wt% of non-fully condensed polyhedral oligomeric silsesquioxane (POSS®Ph-triol) or IL-modified POSS®Ph combined with two different anions chloride (Cl-) or bis-trifluoromethanesulfonimidate (NTf2-). The morphologies and performances these hybrid nanomaterials were characterized and analyzed. Both the epoxy matrix and POSS filler contained ion liquid unit, which is different from the previous epoxy system. Overall, the work is interesting and well organized. It could be considered for acceptance after a minor revision.
1- In Table 1, the value of M = 1.116.04 g mol-1 and M = 1.360.73 g mol-1 contained unnecessary dot. Please check the molecular weight and correct it.
Yes, we have removed the dot from the table.
2-In Table 1, the chemical formula of POSSPh-triol, IL.Cl-g-POSS®Ph and IL.NTf2-g-POSS®Ph needed to be presented in the same font as ILM molecules.
Yes, we have traced with the same font the different structures of POSS as with the molecules of ILMs.
3- In Figure 1, it is better to show Figure 1c and 1f using a same scale bar as Figure 1a and 1b.
We agree with you but unfortunately, we do not have TEM pictures with same sizes for 1c and 1f.
4- The dielectric constant of the hybrid materials should be added for a better evaluation for their possible application in the electrochemistry.
Thank you very much for the comment, the variation curves of the relative permittivity are well displayed on the figure 4 and their different values were discussed in function of the temperature.
Reviewer 4 Report
The New Epoxy Thermosets Organic-Inorganic Hybrid Nanomaterials Derived from Imidazolium Ionic Liquid Monomers and POSS®Ph, of the paper, are very interesting and deserve publication, the quality of the manuscript seems to be rather low and should be significantly improved.
- First and foremost, English quality does not meet the standards of scientific publications.
- The authors should present more information about the nanostructure and classification of the introduction
- Please check the use of acronyms – many of them are not explained, some are exhausting and can be omitted
- Figures quality should be revised.
- Authors should definitely show the relations between permittivity and
Author Response
The New Epoxy Thermosets Organic-Inorganic Hybrid Nanomaterials Derived from Imidazolium Ionic Liquid Monomers and POSS®Ph, of the paper, are very interesting and deserve publication, the quality of the manuscript seems to be rather low and should be significantly improved.
1- First and foremost, English quality does not meet the standards of scientific publications.
2- The authors should present more information about the nanostructure and classification of the introduction
3- Please check the use of acronyms – many of them are not explained, some are exhausting and can be omitted
4- Figures quality should be revised.
Thank you very much for all the suggestions and advices, the entire manuscript has been fully revised by all co-authors and we have tried to correct as many grammatical errors, Figures quality and sentences as possible. Thank you for your understanding.
Round 2
Reviewer 2 Report
The manuscript has been improved, but in my opinion, the authors should add FTIR analysis details to make the experimental design clear and exhaustive.
The mentioned figures S2 and S3 didn't show neither the FTIR spectra nor discussion about which infrared bands were used for the analysis and how the conversion % was calculated from the FTIR data. Please add these details to so as to make the experiment description exhaustive.
4-“FTIR” was mentioned but no spectra or discussions are present.
In fact, FTIR was mentioned in the discussion and the FTIR spectra of the epoxy hybrid O/I networks are presented in the supporting information. To confirm such an assumption, the epoxy conversion was determined by FTIR and the 13C HR-MAS NMR. As shown in Figures S2 and S3 (see supporting information), a high conversion of epoxy-groups was obtained (> 90 %) and the total disappearance of the epoxy peaks (at 34, 47, and 52 ppm in the 13C-NMR spectra) after the curing procedure confirmed that the curing reactions are completed [46, 47]."
Author Response
The manuscript has been improved, but in my opinion, the authors should add FTIR analysis details to make the experimental design clear and exhaustive. The mentioned figures S2 and S3 showed neither the FTIR spectra nor discussion about which infrared bands were used for the analysis and how the conversion % was calculated from the FTIR data. Please add these details to so as to make the experiment description exhaustive.
Thank you very much for your comment and yes, we agree with you. We have added more details about the method used in the manuscript.
In addition, the reaction kinetics, i.e. the quantification of the epoxide groups conversion, for all the reactive systems were investigated by FT-IR spectroscopy following the curing conditions: 3 h at 80 °C, 6 h at 120 °C, and 1h at 200 °C for ILM1/IPD and 1 h at 140 °C followed by 8 h at 190 °C for ILM2/IPD. All the reactive systems were analyzed in the wavelength region of 800 - 1300 cm-1 in order to follow the changes of the adsorption bands at 914 and 1132 cm-1 corresponding to the epoxy groups with reference as the absorption band at 1132 cm-1 (trifluoromethyl (-CF3) of bis(trifluoromethanesulfonyl)imidate counterions (NTf2-) [46], respectively, by using the following equation [28, 45, 46] :
Please see the attachment for the equation
where A0 and At are the ratios between the area of two absorption bands at 914 and 1132 cm-1 (A914/A1132) of the reactive system at the beginning of the reaction (t = 0) and at a given reaction time t. Thus, the epoxy conversion versus the polymerization time is presented in Figure S2a and S2b.
